# Studying SARS-CoV-2 with Fluorescence Microscopy

**DOI:** 10.3390/ijms22126558

**Published:** 2021-06-18

**Authors:** Lidia V. Putlyaeva, Konstantin A. Lukyanov

**Affiliations:** Center of Life Sciences, Skolkovo Institute of Science and Technology, 121205 Moscow, Russia; L.Putlyaeva@skoltech.ru

**Keywords:** COVID-19, SARS-CoV-2, coronavirus, protein labeling, RNA labeling, genetically encoded probes, live cell fluorescence imaging, super-resolution fluorescence microscopy

## Abstract

The COVID-19 pandemic caused by SARS-CoV-2 coronavirus deeply affected the world community. It gave a strong impetus to the development of not only approaches to diagnostics and therapy, but also fundamental research of the molecular biology of this virus. Fluorescence microscopy is a powerful technology enabling detailed investigation of virus–cell interactions in fixed and live samples with high specificity. While spatial resolution of conventional fluorescence microscopy is not sufficient to resolve all virus-related structures, super-resolution fluorescence microscopy can solve this problem. In this paper, we review the use of fluorescence microscopy to study SARS-CoV-2 and related viruses. The prospects for the application of the recently developed advanced methods of fluorescence labeling and microscopy—which in our opinion can provide important information about the molecular biology of SARS-CoV-2—are discussed.

## 1. Introduction

The COVID-19 coronavirus pandemic deeply affected the world community. Millions of deaths, severe health side effects, dramatic economic losses, heavily impacted social life, culture and education—nearly all people in the world have been injured in one way or another. The current situation highlighted the key role of basic studies of animal and human viruses. Only detailed understanding of all steps of interactions between a virus and host cells and immune system would create a solid basis for development of diagnostics, vaccines and drugs.

The current options for treating life-threatening zoonotic coronavirus infections are substantially limited. Although the outbreaks of SARS in 2003 and MERS-CoV in 2012 triggered extensive research efforts, there are currently no drugs that can effectively treat any zoonotic coronavirus. Obviously, none of the inhibitors of viral enzymes can provide absolutely reliable protection because of the ability of most viruses of rapid mutation under the pressure of natural selection. To prevent such a rapid adaptation of viruses to a particular drug, it is essential to exploit the cumulative effect of drugs on several proteins at once. Therefore, an extensive search for protein inhibitors from various functional groups (proteases, polymerases, structural proteins) appears to be highly relevant. The search for new treatment strategies is inextricably linked with both fundamental research on molecular mechanisms of virus life cycle and creation of new screening platforms for efficient testing of viral protein inhibitors.

Fluorescence microscopy is a relatively simple and robust technology, which is broadly used in virtually all fields of biology including virology. Importantly, it provides extremely high sensitivity (down to single molecules) and specificity due to antibody staining or genetically encoded tags. Another key advantage of fluorescence microscopy is a possibility to work with live cells enabling studying structures and processes in dynamics in real time. Spatial resolution of conventional fluorescence microscopy is not very high (200–300 nm). At the same time, recent advances in super-resolution fluorescence microscopy (nanoscopy) ensure imaging in nanometer scale. Fluorescence microscopy has been used for investigating viruses and their interactions in cellular systems [1,2,3,4,5]. This review focuses on applications of fluorescence microscopy to study SARS-CoV-2 and related viruses.

## 2. Overview of SARS-CoV-2 Life Cycle

An outline of the main stages of SARS-CoV-2 life cycle is shown in Figure 1 [6].

Attachment of the virion to the host cell is initiated by the interaction between the viral protein S (spike) and its receptor, angiotensin converting enzyme 2 (ACE2). S protein of coronaviruses contains two important functional subunits, S1 and S2, that are important determinants of tissue tropism and host range. After binding of the S protein to ACE2, protein S is cleaved by transmembrane serine protease 2 (TMPRSS2) at the plasma membrane or by proteases furin or cathepsin in the late endosomes or lysosomes. Then, the S2 subunit dissociates and S1-mediated fusion of the viral and cell membranes occurs resulting in entry of viral genomic RNA into the cytosol [7].

The next stage in the coronavirus life cycle is the translation of a so-called replicase gene from the viral genomic RNA. The replicase gene encodes two large ORFs, rep1a and rep1b, which express two polyproteins, pp1a and pp1ab. The pp1a and pp1ab polyproteins contain nsp (non-structural proteins) 1–11 and 1–16, which are subsequently cleaved into separate nsp [8]. Coronaviruses encode two proteases that cleave replicase polyproteins: the papain-like protease (PLpro), encoded by nsp3, and the serine proteinase Mpro (also named 3CLpro) encoded by nsp5. PLpro cuts the nsp1/2, nsp2/3 and nsp3/4 boundaries, while Mpro is responsible for cleavage of the remaining 12 proteins.

Mpro is a small, soluble protein that can be expressed in *E. coli* in a fully functional form, that makes it relatively easy to study and screen for specific inhibitors [9]. On the contrary, understanding the functioning of PLpro faces a number of serious difficulties. The protease domain PLpro is part of a large polydomain protein nsp3 of ~2000 amino acids with four transmembrane domains. Expression of full-length nsp3 is a challenging technical task (for example, in a recent large-scale study, all proteins of the virus were expressed, except for nsp3 and nsp16 [10]).

Other nsps constitute enzymes and proteins that are important for RNA replication, for example, nsp12 gene encodes RNA-dependent RNA polymerase (RdRp); nsp13 gene—RNA helicase and RNA-5′-triphosphatase; nsp14 gene—exoribonuclease (ExoN), which affects the accuracy of replication and activity of N7-methyltransferase; and nsp16 encodes 2′-O-methyltransferase activity.

Following the translation of nsp proteins, viral replicase complexes are assembled and viral RNA is synthesized within double membrane vesicles (DMVs) formed from ER membranes. During synthesis, both genomic and subgenomic RNAs are formed. Subgenomic RNAs serve as messenger RNAs for structural (S, M, N, E) and accessory genes, which are located closer to the 3′-end of the replicase polyproteins.

Structural proteins S, E and M are translated and inserted into the ER. These proteins move along the secretory pathway to the intermediate compartment between the ER and the Golgi apparatus, ERGIC [11]. Protein M directs most of the protein–protein interactions in the assembly of coronaviruses, but expression of the E protein is also required for the formation of virus-like particles (VLPs) [12]. It was shown that the interaction of proteins S and E is physically possible, which can be due to the interaction of homologous cysteine sequences at the C-termini of proteins [13]. The data was obtained using a tandem affinity purification system in combination with mass spectrometry (MS; TAP-MS), but have not been confirmed in living cells. Protein M also binds to the nucleocapsid (the CTD domain of protein N), and this interaction contributes to the completion of virion assembly. However, it is unclear how the nucleocapsid in complex with the virion RNA enters the ERGIC to interact with the M protein and gets inside the viral envelope formed from M, E and S. The packaging of the full-length positive-sense genomic RNA of the virus occurs due to the 151-nucleotide packing signal found in the *nsp12* coding sequence [14]. After assembly, virions are transported to the cell surface in vesicles and released through exocytosis.

## 3. Fluorescence Microscopy-Based Studies of Viral Life Cycle

Fluorescence microscopy is a classical approach to identify virus infection which was used in studies of different viruses, e.g., previous coronavirus infections (SARS-CoV and MERS-CoV) [15]. Fluorescence methods allow visualization of viruses and their specific components (proteins and genome) as well as target structures and events in the host cells.

Virus entry is a multistep process. It initiates when the virus attaches to the host cell and ends when the viral contents reach the cytosol. In a series of papers, confocal fluorescence microscopy was used to investigate the endocytic pathway by the virus. It was found that SARS-CoV enters cells through pH- and receptor-dependent endocytosis. In the two most profound works, the authors labeled the SARS-CoV functional receptor ACE2 with GFP by stable transfection of human embryonic kidney 293 (HEK293) cells. Receptor recycling was then tracked after the cells were treated with pseudoviruses or the spike protein, a membrane component of SARS-CoV that mediates membrane fusion and is required for viral entry [16]. The recent study revealed the role of a secretory form of ACE2 receptor (sACE2) in SARS-CoV-2 infection [17]. Additionally, high-resolution cryo-electron microscopy (cryo-EM), atomic force microscopy and confocal microscopy allowed Yang et al. to describe kinetic and thermodynamic properties of this binding pocket of S-glycoprotein with the ACE2 receptor [18]. Using cryo-EM, Schoof et al. demonstrated that a single domain antibody Nb6 stabilizes spike protein in a conformation where its binding domain is inaccessible for interactions with ACE2 [19].

Some key aspects of SARS-CoV-2 cell entry were clarified in 2020 due to consolidation of the efforts of scientists around the world to find new potent inhibitors of SARS-CoV-2. In [20], it was confirmed that human ACE2 is the receptor for SARS-CoV-2. The authors also demonstrated that SARS-CoV-2 enters HEK293/hACE2 cells mainly through endocytosis with critical importance of PIKfyve, TPC2 and cathepsin L for the virus entry. Fluorescence microscopy was also used to demonstrate cell–cell fusion and syncytium formation mediated by SARS-CoV-2 S protein in HEK293 cells. In another work, it was demonstrated the potent inhibitory effects of Apilimod and Vacuolin-1, small-molecule inhibitors of PIKfyv, on content release and infection by chimeric vesicular stomatitis virus (VSV) containing the envelope proteins of Zaire ebolavirus or SARS-CoV-2 [21]. The authors also describe potent inhibition of SARS-CoV-2 by Apilimod. Another group showed that SARS-CoV and CoV-2 requires the cell surface heparan sulfate (HS) as an assisting cofactor for virus entry [22]. Confocal microscopy analysis showed that the actin network is required for spike-mediated viral entry and abrogated by inhibitors Sunitinib and BNTX.

For many viruses, tissue tropism is determined by the availability of virus receptors and entry cofactors on the surface of host cells. Cantuti-Castelvetri et al. found that neuropilin-1 (NRP1), known to bind furin-cleaved substrates, significantly potentiates SARS-CoV-2 infectivity, an effect blocked by a monoclonal blocking antibody against NRP1. NRP1 is abundantly expressed in the respiratory and olfactory epithelium, with highest expression in endothelial and epithelial cells. Results were obtained both from SARS-CoV-2 pseudotyped particles and SARS-CoV-2 viruses isolated from COVID-19 patients from the Helsinki University Hospital [23]. In another work, Lv et al. demonstrated that primary M1 alveolar macrophages (AM) isolated from murine bronchoalveolar lavage fluid took up SARS-CoV-2 virus more effectively than M2 AM [24]. RNAscope assay with subsequent confocal microscopy imaging revealed that that viral RNA appeared in M1 AMs much faster than in M2 AMs, and anti-NP and anti-Rab7 staining demonstrated that SARS-CoV-2 can effectively replicate in M1 AMs because of the lower endosomal pH in M1 AMs.

An important goal is not only to visualize virus entry events, but to develop new cell and tissue models of SARS-CoV-2 infection. Zhou et al. established intestinal organoid culture using crypts isolated from the intestines of *Rhinolophus sinicus* bats to create accessible in vitro model that can faithfully represent native bat cells, and compared the results with human intestinal organoids [25]. Confocal imaging of the infected human enteroids after co-staining of viral NP and villin, a marker of human enterocytes, demonstrated that the most infected cells in the enteroids were villin-positive, indicating that enterocytes are the major target cell of SARS-CoV-2. Additionally, Zaeck et al. combined the light sheet fluorescence microscopy and tissue optical clearing to make the 3D overview of SARS-CoV-2 infection in the ferret model [26]. Modified iDisco protocol allowed to achieve optically cleared ferret nasal turbinates and lung tissues and then to perform immunostaining with polyclonal serum and monoclonal antibodies against N protein. The study confirmed SARS-CoV-2 preferential infection of the upper respiratory tract in the ferret and revealed a distinct oligofocal infection pattern of infection in nasal turbinates (Figure 2).

Micro-environment on the cell surface also can determine the effectiveness of viral entry into the host cell. The confocal fluorescence images showed that the ectodomain (S1188HA) of SARS-CoV S protein could associate with lipid rafts after binding to its receptor, and colocalize with raft-resident marker ganglioside GM1 [27]. These data were fully confirmed for SARS-CoV-2 in 2021: it was shown that infection of SARS-CoV-2 pseudoviruses is independent of dynamin, clathrin, caveolin and endophilin A2, as well as macropinocytosis, but it is cholesterol-rich lipid raft dependent [28]. In this work, fluorescence microscopy was used to demonstrate the GFP-encoding pseudovuruses’s infectivity of Vero E6 cells in the presence/absence of inhibitors or siRNA targeting each endocytic pathway. In contrast, Bayati et al. showed that SARS-CoV-2 infects cells precisely via clathrin-mediated endocytosis [29]. Using purified spike glycoprotein and lentivirus pseudotyped with spike glycoprotein and marked with GFP, it was demonstrated that after engagement with the plasma membrane, SARS-CoV-2 undergoes rapid, clathrin-mediated endocytosis. In this paper authors used two drugs that are known to block clathrin-mediated endocytosis, Dynasore and Pitstop 2, while Li et al. [28] used Chlorpromazine and si-Clathrin. Considering this controversial data, the precise mechanism of SARS-CoV-2 endocytosis host cells remains not fully clear.

Another important step of virus entry is transport of viral particles via endosomes into the cell and evaluation of intracellular transport mechanisms that may cause different infectiousness of cells. Nicotinic acid adenine dinucleotide phosphate (NAADP) is a second messenger that releases Ca^2+^ from acidic organelles through the activation of two-pore channels (TPCs) to regulate endolysosomal trafficking events. Gunaratne et al. found that knockdown of JPT2 diminished NAADP-mediated Ca^2+^ signals from endosomes and lysosomes and the ability of a SARS-CoV-2 pseudocoronavirus to infect cells, a process that depends on two-pore channels (TPC) activity [30]. It was established that JPT2 is a component of the NAADP receptor complex that is essential for TPC-dependent Ca^2+^ signaling and control of coronaviral entry. The importance of NAADP regulation was also confirmed for MERS-CoV: it was shown that Ca^2+^-permeable channels within the endolysosomal system regulate both the luminal environment and trafficking events of MERS-CoV. Knockdown of endogenous TPCs, targets for the Ca^2+^ mobilizing second messenger NAADP, impaired infectivity in a MERS-CoV spike pseudovirus particle translocation assay. Colocalization of MERS-CoV spike protein with endolysosomal ion channel positive structures (TPC1-GFP, TPC2-GFP and GFP-TRPML1) was showed by immunostaining and subsequent visualization by confocal microscopy [31].

The next step of the virus life cycle is the release of viral RNA into the host cell, translation of the polyproteins and RNA replicase–transcriptase complex formation. This process is supported by an elaborate virus-induced network of transformed ER membranes known as the viral replication organelle (RO). Double-membrane vesicles (DMVs) are the RO’s most abundant component and the central hubs for viral RNA synthesis. More than 15 years ago it was found that the SARS-CoV 3a protein is important for the viral life cycle and is found to localize within the Golgi complex and plasma membrane of the infected cells and is an integral membrane protein. Co-labeling of nsp-3 or nsp-8 responsible for RNA replication showed its close association with the dsRNA [32,33]. These data were confirmed for SARS-CoV-2: Wolff et al. used cryo-EM to analyze the structure of coronavirus-induced ROs and demonstrated that the coronavirus transmembrane protein nsp3 is a component of the pore complex [34]. They proposed a new model of release of newly synthesized viral RNA and RNP formation: specific replicase subunits may associate with the pore complex to guide the newly synthesized +RNA toward it. On the cytosolic side, part of RNAs would then be used for translation, together with the much smaller, though much more abundant, subgenomic mRNAs. Genome-containing RNP complexes would travel to the membranes where the viral envelope proteins accumulate and engage in the assembly of progeny virions. The virions bud into single-membrane compartments, typically derived from ERGIC, and travel along the secretory pathway to be released into extracellular space. Fluorescent protein EGFP was used to mark nsp3 protein involvement in pore complex formation.

As mentioned above, the virus assembly takes place in ERGIC. Many studies have been devoted to exploring the minimal requirement for the assembly of virus-like particles (VLPs) for the coronaviruses SARS-CoV and SARS-CoV-2. Siu et al. showed that M, E and N structural proteins are required for efficient production of SARS-CoV VLPs [35]. The investigators engineered plasmid constructs that allow expression of viral proteins in fusion with fluorescent proteins in order to visualize in real time the assembly, trafficking and release of SARS-CoV VLPs. Recently, Boson et al. combined biochemical and imaging assays in infected versus transfected cells and showed that E and M proteins regulate intracellular trafficking of S as well as its intracellular processing. Indeed, the imaging data revealed that S is relocalized at ERGIC or Golgi compartments upon coexpression with E or M (Figure 3) [36].

## 4. Influence of Viral Proteins on the Host Cell

The function of most viral proteins was evaluated without the use of fluorescence microscopy, but some important issues were clarified using this technique. Some viral proteins are suspected of being ‘viroporins’ [37], i.e., viral proteins that can form ion channels, and may play a role in several processes, including virus replication and pathogenesis. Earlier it was shown that some coronaviruses (MERS-CoV, HCoV-OC43, HCoV-229E) are characterized by the presence of two types of viroporins, while SARS-CoV encodes three ones: proteins 3a, E and 8a [38]. It was demonstrated that only 3a and E proteins were clearly involved in SARS-CoV replication and virulence, while viroporin 8a had only a minor effect on these activities. Confocal microscopy showed that 3a protein did not localize at any of the main intracellular calcium storage locations, whereas E protein localizes to the ERGIC. Additionally, it was shown that oligomerization of SARS 3a causes necrotic cell death, caspase-1 activation and lysosomal damage during SARS-CoV infection [39]. Using cryo-EM and fluorescent ion flux assays Kern et al. also confirmed these data for SARS-CoV-2 and demonstrated that protein 3a forms a functional ion channel that may promote COVID-19 pathogenesis [40]. Indirect immunofluorescence microscopy analysis showed that the distribution of SARS-CoV E protein overlapped most extensively with that of the cis-Golgi marker, GM130 [41]. E protein overlap was less extensive with the ERGIC marker, ERGIC53, and the trans-Golgi marker, p230, which suggests that the SARS-CoV E protein is targeted to the cis-Golgi region when expressed in HeLa cells. It was also demonstrated that the C-end of the SARS-CoV E protein contains Golgi complex-targeting information. Additionally, Cabrera-Garcia et al. confirmed these data for SARS-CoV-2: they showed that the wild-type SARS-CoV-2 E protein fused with fluorescent carboxyl-terminal tag (mKate2), can be trafficked to intracellular organelles, lysosomes and ERGIC [42]. Their novel vectors for E protein expression enabled authors to study its dual functions: the ion channel activity of SARS-CoV-2 E when it is expressed in the plasma membrane, and its effect on proton homeostasis when located to intracellular organelles. Collected data indicated that E protein forms a cation channel, viroporin, that is modulated by changes of pH.

Yuan et al. observed that transient expression of ORF7a of SARS-CoV protein fused with myc or GFP tags at its N or C terminus inhibited cell growth and prevented BrdU incorporation in different cultural cells, suggesting that ORF7a expression can regulate cell cycle progression [43].

Comprehensive evaluation of the interactions between SARS-CoV-2 proteins and human RNA species showed that ten viral proteins form highly specific interactions with mRNAs or noncoding RNAs (ncRNAs), including those involved in progressive steps of host cell protein production [44]. It was shown that nsp16 binds to the mRNA recognition domains of the U1 and U2 splicing RNAs and acts to suppress global mRNA splicing upon SARS-CoV-2 infection. To test this, nsp16 was co-expressed with a splicing reporter derived from IRF7 (an exon-intron-exon minigene) fused to GFP in human cells. Additionally, authors explored whether the leader sequence protects viral mRNAs from translational inhibition with nsp1 by fusing the viral leader sequence to the 5′ end of GFP or mCherry reporter genes. They evaluated that nsp1 binds to 18S ribosomal RNA in the mRNA entry channel of the ribosome and leads to global inhibition of mRNA translation upon infection. Finally, nsp8 and nsp9 bind to the 7SL RNA in the signal recognition particle and interfere with protein trafficking to the cell membrane upon infection, which was shown by immunostaining.

## 5. RNA Tracking

Over the last few decades, fluorescence-based methods have been developed to label the RNA in cellulo with high specificity and sensitivity. While biochemical techniques based on total RNA extraction from pools of cells and viruses provide information on the average behavior of a population, fluorescence microscopy enables direct visualization of single RNA molecules within single cells and viruses [45].

Recent data demonstrated that SARS-CoV-2 N undergoes RNA-dependent phase separation. The authors generated a variant N protein with an N-terminal cysteine for specific labeling and mixed Cy5-labeled full-length N protein with a nonspecific 17-mer ssRNA labeled with fluorescein (6-FAM). After mixing formation of spherical phase-separated condensates containing both components was observed. Additionally, full bleaching and partial bleaching of N + RNA condensates indicated that the N protein is very slowly exchanged with the soluble pool and within the structures, with recovery of only ~12% fluorescence intensity in 2 min after bleaching [46]. Authors also made representative STED super-resolution images of three-component condensates demonstrating that the removal of the C-terminal dimerization domain and C-terminal IDR of N protein completely eliminated N + M phase separation, that supports the idea that the N protein’s C-terminal region interacts with M, and that this interaction can promote the assembly of condensates in the absence of RNA.

The anti-dsRNA monoclonal antibody is the gold standard in dsRNA detection and still actively used. Using cycloheximide and puromycin, van den Worm et al. studied the effect of translation inhibition on the RNA synthesis of SARS-CoV and MHV. Rabbit antisera to nsp4 and a mouse monoclonal antibody labeling double-stranded RNA allowed authors to perform dual-immunofluorescence microscopy and reveal that inhibitors cycloheximide and puromycin prevent the usual exponential increase in viral RNA synthesis [47].

Unlike immunostaining, PCR or sequencing methods, fluorescence in situ hybridization (FISH) offers the capacity to directly and specifically visualize viral RNA in single cells. In situ hybridization (ISH) has been used in 30 out of 136 studies on detection methods of SARS-CoV-2 in tissues to March 2021. Chromogenic in situ hybridization (CISH) was used in the majority of studies, but fluorescent ISH (FISH) can also be used [48].

During 2020, the “classical” FISH approach undergoes significant adaptation to the SARS-CoV-2 needs. Rensen et al. designed and validated previously reported smiFISH probes against the positive and negative RNA strands of SARS-CoV-2 and named them CoronaFISH. CoronaFISH provides a flexible, cost-efficient and versatile platform for studying SARS-CoV-2 replication at the level of single cells in culture or in tissue and can potentially be employed for drug screening and diagnosis. Additionally, it is possible to adopt CoronaFISH for electron microscopy in situ hybridization (EM-ISH) [49].

Recently, Liu et al. established a multiplex FISH (mFISH) to detect positive-sense SARS-CoV-2 RNA in formalin-fixed paraffin-embedded (FFPE) specimens and showed that, consistent with the RNAscope (an approach which was previously developed for detection of various high-consequence viruses including Ebola virus (EBOV), Marburg virus (MARV), Lassa virus (LASV), etc. in FFPE animal tissues) ISH results, positive-sense viral RNA was widely distributed in the cytoplasm, whereas negative-sense RNA (replicative intermediate) was confined to perinuclear inclusion bodies. They also developed a dual staining assay using IHC and ISH to detect SARS-CoV-2 S protein and viral RNA in the same FFPE section and clearly demonstrated that SARS-CoV-2 antigen was detected along with positive-sense RNA in the cytoplasm of most of the infected, but not in uninfected, cells [50]. Another optimized variant of the FISH method for SARS-CoV-2 RNA detection is Immuno-RNA-FISH, which represents a combination of the RNA-FISH coupled with immunofluorescence analysis of cell lines and 3D tissues (fully differentiated human airway epithelium). The method relies on the concept of a hybridization chain reaction initiated by appropriate probe localization and on the use of split-initiator probes to begin amplification of the signal by fluorescently labeled amplifiers results in minimal-to-no background fluorescence when observed using a confocal microscope [51].

## 6. SARS-CoV-2 Drug Screening and Inhibitor Testing

The field of drug discovery and development for COVID-19 antivirals requires tools and reagents to study the viral mechanisms of infection in order to identify targets for therapeutic intervention. One important direction is evaluation of the appropriate antibodies against SARS-CoV-2 proteins among the existing. Ogando et al. demonstrated that the most nsps of SARS-CoV-2 had a strong cross-reactivity with SARS-CoV nsps (nsp3, nsp4, nsp5, nsp8, nsp9, nsp13, nsp15) with the exception of polyclonal nsp6 rabbit antiserum, and for structural proteins: N protein, C-terminal peptide of the SARS-CoV M protein [52]. Another direction is to identify and validate inhibitors of the SARS-CoV-2 spike and ACE2 receptor binding in human cells. To this end, Gorshkov et al. developed versatile nanoparticle probes consisting of spike subunits conjugated to quantum dots (QDs) [53]. This probe is capable of engaging in energy transfer quenching with ACE2-conjugated gold nanoparticles to enable monitoring of the binding event in solution. Neutralizing antibodies and recombinant human ACE2 blocked quenching, demonstrating a specific binding interaction.

A prominent goal to prevent cellular entry of SARS-CoV-2 is to find an inhibitor to transmembrane serine protease (TMPRSS2). Chen et al. developed a fluorescence resonance energy transfer (FRET)-based platform for effectively screening inhibitors against TMPRSS2 protease activity [54]. The disruption of FRET between green and red fluorescent proteins conjugated with the substrate peptide, which corresponds to the cleavage site of SARS-CoV-2 spike protein, was measured to determine the enzymatic activity of TMPRSS2; Flupirtine was found to be the most promising inhibitor among other FDA-approved drugs.

Another group of approaches is development and validation of a reporter optimized to detect SARS-CoV-2 protease activity. Froggatt et al. published the new reporter based on a FlipGFP, which fluoresces only after cleavage by 3CLpro [55]. 3CLpro is an attractive target for antiviral therapeutics, as it is essential for processing newly translated viral proteins. This experimentally optimized reporter assay allows for antiviral drug screening in human cell culture at biosafety level 2 (BSL2) with high-throughput compatible protocols. This reporter was also used to test the inhibition of SARS-CoV-2 3CLpro with a known coronavirus 3CLpro inhibitor, GC376, and then validated the correlation between reporter inhibition and inhibition of SARS-CoV-2 replication. Recently, Xia et al. developed the cell-based FlipGFP assay that is suitable for quantifying the intracellular enzymatic inhibition potency of PLpro inhibitors in the BSL-2 setting [56]. Using FlipGFP-PLpro assay, two promising compounds that inhibited SARS-CoV-2 replication in a cell model in micromolar concentrations were found. Another assay to identify coronavirus 3CLpro inhibitors was published by Resnick et al. [57]. This assay is based on rescuing protease-mediated cytotoxicity and does not require live virus. By enabling the facile testing of compounds across a range of 15 distantly related coronavirus 3CLpro enzymes, compounds with broad 3CLpro inhibitory activity were identified.

Additionally, some efforts were made to implement machine learning to distinguish SARS-CoV-2 in clinical samples between other viruses. In the work of Shiaelis et al. viral particles were labeled with Cy3 or Atto647N fluorophores and imaged using the internal reflection fluorescence (TIRF) microscopy [58]. The trained network demonstrated the efficiency of distinguishing between SARS-CoV-2, seasonal hCoVs (OC43, HKU1 or NL63) or human influenza A samples with accuracy ~70%.

## 7. Recombinant SARS-CoV-2 Expressing Reporter Genes

Laboratory studies of SARS-CoV-2 allow the use of additional methods to check for the presence of the virus in infected cells. The ability to generate reporter-expressing recombinant viruses represents a powerful tool to address important questions about the pathogenesis of viral infection as well as for the easy and rapid identification of candidate antiviral drug [59].

To create a competent system for BSL-2 studies, Plescia et al. studied formation of virus-like particles (VLPs) by the four structural proteins of SARS-CoV-2 (namely, M, S, E and N) expressed in human cells [60]. It allowed authors to confirm that SARS-CoV-2 VLP production is driven by M coexpression with additional viral proteins. Moreover, fluorescently labeled VLPs, based on the coexpression of S-GFP with M, N and E in HEK293 cells, were able to entry the cells demonstrated colocalization with endocytic pathway markers mCherry-Rab5 (early endosomes) and mCherry-LAMP1 (lysosomes).

Several groups have described the ability to generate recombinant SARS-CoV-2 (rSARS-CoV-2) expressing different reporter genes using reverse genetics technology. Hou et al. utilized a reverse genetics system to generate a GFP-tagged SARS-CoV-2 and revealed that SARS-CoV-2 shows a gradient infectivity from the proximal to distal respiratory tract and that ciliated airway cells and AT-2 cells are primary targets for SARS-CoV-2 infection [61]. In another study, mNeonGreen fluorescent protein was introduced into ORF7 of the SARS-CoV-2 genome [62] similarly to how it was done earlier for SARS-CoV [63]. Importantly, the presence of mNeonGreen did not affect virus replication kinetics; the level of fluorescent signal in the cells was found to correlate with the initial multiplicities of infection. A robust fluorescence readout enabled authors to assess the antiviral activity of interferon in this model. The same engineering strategy was used to construct fully infectious recombinant SARS-CoV-2-Nluc virus expressing NanoLuc luciferase, which can be used to measure neutralizing antibody in patient sera as well as for high-throughput antiviral drug screening [64]. Authors noted that due to the amplifying nature of Nluc enzyme, the SARS-CoV-2-Nluc assay has a greater dynamic range and higher sensitivity than the fluorescent mNeonGreen-tagged virus assay.

Thao et al. recently reported a yeast-based synthetic genomics platform for fast and reliable genetical reconstruction of diverse RNA viruses, including members of the Coronaviridae (e.g., SARS-CoV-2) [65]. Transformation-associated recombination (TAR) cloning was used to create SARS-CoV-2-GFP constructs and obtain correctly assembled molecular clones. It is worth noting that the yeast-platform-produced SARS-CoV-2 should be fully characterized for its biological properties (e.g., replication kinetics) in comparison with its original clinical isolate.

Reverse genetics protocols often require complex in vitro engineering procedures before transfecting cells. Chiem et al. described the generation and characterization of replication-competent rSARS-CoV-2 expressing fluorescent Venus or mCherry or bioluminescent Nluc reporter genes using recently described bacterial artificial chromosome (BAC)-based reverse genetics approach [59]. The results demonstrated that cells infected with rSARS-CoV-2-Venus or -mCherry express the corresponding reporter genes and that viral infections can be easily detected by fluorescence in 24–72 h postinfection; or rSARS-CoV-2-Nluc—by luminescence in 48 h.

## 8. Perspectives

Fluorescence microscopy-based studies of SARS-CoV-2 and related viruses discussed above are summarized in Table 1. Notably, mainly old labeling and imaging techniques were used up to date, whereas cutting-edge methods are still waiting to be applied to this subject. Despite the fact that SARS-CoV-2 has been extremely thoroughly studied over the past year, many important questions remain unclear. What is the molecular mechanism of DMV formation from ER membranes? What is the protein composition of the outer and inner membranes of DMVs? How does the pore complex in DMVs discriminate between positive- and negative-sense RNAs? Where do viral structural proteins begin to interact with each other and with genomic RNA? These and other topics can potentially be addressed with advanced methods of fluorescence labeling and microscopy. In our opinion, the following directions can provide important insights into molecular biology of SARS-CoV-2 and related viruses.

### 8.1. Early Labeling and Tracking of Viral Proteins

The classical method for imaging of target proteins in living cells is the use of fusion proteins with green fluorescent protein GFP or GFP-like proteins of various colors. However, maturation of the chromophore group in these fluorescent proteins requires a considerable time (at least tens of minutes) (Figure 4A). This represents a significant problem for the analysis of the synthesis and transport of membrane proteins: the detected fluorescent signal mainly corresponds to the late stages of transport, while the early stages remain invisible. This problem can be addressed using different types of genetically encoded tags as discussed below.

In a new method, called KECs (K/E coils), the fluorescent protein and the target protein are not part of a single polypeptide chain, but are linked through reversible specific heterodimerization of alpha-helices (one helix is fused with a fluorescent protein, the other with the target protein, Figure 4B) [66]. As a result, a pre-expressed and matured fluorescent protein in the cytosol immediately binds to the newly synthesized target protein and reveals its localization (the method is applicable only for proteins with distinct localization, for example, cytoskeletal and membrane proteins). Using the SSTR3 somatostatin receptor as an example, it was demonstrated that KECs-labeling allows the detection of membrane proteins at much earlier stages of their intracellular transport than classical labeling with GFP. Another important advantage of KECs is that the constant exchange of the pool of fluorescent proteins (between those bound to the target protein and freely diffusing in the cytosol) makes it possible to achieve very high apparent photostability of the fluorescent signal in some imaging modalities (illumination of only a part of the cell in confocal and TIRF microscopy), which is important for long-term imaging or obtaining high-resolution images of selected cell parts. An obvious drawback of KECs method is the presence of background fluorescence of unbound fluorescent protein in the cytosol.

The latter problem is avoided in labeling with fluorogen-activating proteins (FAPs) that bind non-fluorescent fluorogenic dyes to form fluorescent complexes (Figure 4C,D). It was discovered that a natural protein UnaG from eel binds endogenous bilirubin with a high affinity and displays bright green fluorescence [67]. Thus, UnaG can be used similarly to GFP, but UnaG does not require a long time for its maturation (Figure 4C). There are also a number of artificially engineered FAPs of different origin that recognize exogenously added cell-permeable fluorogens. Since free fluorogens are non-fluorescent, there is no need for washing steps, and the labeling procedure is simple and fast (Figure 4D). For example, DiB (Dye in Blc) tags reversibly bind a chemical fluorogen enabling high-photostability imaging of target proteins [68,69]. Moreover, continuous exchange between bound and free fluorogen molecules provides an efficient way for super-resolution fluorescence microscopy in both single molecule detection and stimulated emission depletion modalities.

### 8.2. Aptamer-Based RNA Labeling

A powerful approach of target RNA labeling in live cells was developed during the last decade. It is based on RNA aptamers specifically binding cell-permeable fluorogenic dyes, which are non-fluorescent in solution but become brightly fluorescent (>1000-fold fluorescence increase) in a complex with the cognate aptamer [70]. While the early versions of aptamers (e.g., Spinach) suffered from low stability and high thermal and ion sensitivity, recent improvements of both aptamers (e.g., Broccoli) and fluorogens resulted in robust labeling of target RNAs in mammalian cells [71]. A possibility of imaging single RNA molecules inside the Cos-7 cells was demonstrated with the fluorogenic Mango II arrays [72]. The array was found to ensure higher signal to noise ratio than fluorescent protein-based markers and can be used for prolonged tracking of single RNA molecules.

These light-up aptamers can potentially be used to follow the fate of model coronavirus positive- or negative-sense RNAs. Importantly, due to the membrane-permeable nature of the fluorogens, this approach is applicable to any cell compartments including DMVs and ERGIC. In contrast, fluorescent protein-based methods, such as Pepper aptamer [73], are restricted to RNA species contacted with cytosol.

Another promising method of target RNA labeling is based on the stabilization of a rapidly degrading fluorescent protein in the presence of an RNA aptamer Pepper [73]. This system allows visualization of the target mRNA carrying multiple copies of the Pepper aptamer in the 3′-non-coding region in living mammalian cells, down to the level of single molecules. However, this method seems to be restricted to RNA species directly contacted with cytosol or nucleoplasm, whereas RNA molecules located in vesicles would remain unstained.

### 8.3. Monitoring Cell Physiology with Fluorescent Sensors

Virus life cycle relies on many cellular molecular machineries and signaling events. For example, blocking of NAADP signaling inhibits endocytic entry of SARS-CoV-2 [29]. In turn, coronavirus and even its proteins alone can change some aspects of cell physiology before general toxicity. The use of various genetically encoded fluorescent sensors can shed light on such processes. In particular, Ca^2+^ sensors, e.g., of GCaMP series [74], should be used to reveal possible changes of calcium ion concentration associated with different steps of SARS-CoV-2 propagation. Another important signaling molecule potentially involved in the virus life cycle is hydrogen peroxide, which can be monitored with HyPer variants [75,76].

Practically all stages of the coronavirus intracellular pathway are based on membranes. It is known for some viruses that viral proteins directly affect membrane composition and shape. For example, Hepatitis C virus non-structural protein 5A interacts with the host’s phosphatidylinositol 4-kinase IIIα (PI4KA) that results in the attachment of ER to mitochondria and fragmentation of the latter [77]. Thus, it is of high interest to follow the dynamics of various polyphosphoinositides (PPIn), which act as second messengers and shape membranes, during SARS-CoV-2 infection. Various genetically encoded fluorescent sensors with a translocation-based readout can be used for this task [78].

In addition, label-free fluorescence lifetime microscopy (FLIM) makes it possible to measure metabolic status of a live cell by assessing fluorescence of endogenous redox cofactors—NAD(P)H and FAD/FMN [79]. This method provides a sensitive way to detect real-time changes of cell metabolism in various biological processes including virus infection or expression of viral proteins [4].

### 8.4. Correlative Fluorescence and Electron Microscopy

Fluorescence microscopy enables highly specific labeling of target proteins but possesses too low resolution to reveal details of most virus-associated structures. At the same time, it is difficult to identify target protein in electron microscopy. Thus, correlative fluorescence and electron microscopy represents a promising way of nanometer-precision visualization of structures pre-identified in fluorescence images [80]. In addition, singlet oxygen producing fluorescent flavoprotein miniSOG and its improved variants, such as SOPP3 [81], can be used for light-induced polymerization of diaminobenzidine into an osmiophilic polymer, which acts as a contrast for electron microscopy [82]. These methods can potentially be used to localize SARS-CoV-2 proteins within DMVs and other membrane structures.

### 8.5. Super-Resolution Fluorescence Microscopy

Another way to improve resolution to resolve virus-related structures is super-resolution fluorescence microscopy (SRFM) that allows to break the diffraction limit of optical microscopy [83,84]. In 2014, the Nobel Prize in Chemistry was awarded to E. Betzig, S. Hell and W. Moerner for two main strategies of SRFM—stimulated emission depletion (STED) and single molecule localization microscopy (SMLM). STED microscopy is based on scanning with two collinear lasers; the first laser excites fluorescence, while the second very high intensity red-shifted laser beam shaped as a doughnut depletes fluorescence around the small central spot. The resulting fluorescence spot is very small (usually 40–60 nm for biological samples). SMLM relies on calculation of fluorophore positions from the image of single molecules and allows to achieve an even higher resolution, down to 10–20 nm. A recent breakthrough in the field of SRFM was MINFLUX—a method that combines scanning with a doughnut-shaped laser with single-molecule detection [85,86]. MINFLUX achieves resolution in the range of units of nanometers and also makes it possible to track fast movements of single molecules in the sub-millisecond time scale and nanometer precision.

While SRFM would ensure to visualize specific viral proteins with very high spatial resolution and practical compatibility with live-cell imaging [87], quite a few groups implemented it in the SARS-CoV-2 studies to date. Stebbing et al. demonstrated that baricitinib, approved as a treatment for adult rheumatoid arthritis, can be a potential therapeutic through the inhibition of viral endocytosis or blockade of JAK1/2 kinase [88]. Super-resolution dSTORM microscopy of infected liver spheroids stained for nucleocapsid protein demonstrated that baricitinib efficiently blocked SARS-CoV-2 entry. We believe that further application of various SRFM methods (in particular, MINFLUX) for fixed and live objects would reveal detailed picture of all steps of SARS-CoV-2 interactions with a host cell.

### 8.6. Expansion Microscopy

Expansion microscopy (ExM) is a recently developed technique that enables nanoscale-resolution imaging of preserved cells and tissues on conventional diffraction-limited microscopes via isotropic physical expansion of the specimens before imaging [89]. To date, many ExM protocols have been implemented on various issues: proExM, iExM, exFISH, etc. [90]; also, several protocols for combination of expansion and super-resolution microscopy were proposed: for example Structured Illumination Microscopy (SIM) [91], STED [92] and SMLM [93]. ExM can also be used in detecting disease-causing pathogens like parasites, bacteria, viruses and fungi in particular because of its applicability on conventional microscopes or developing mini-microscopes [94]. Recently Gao et al. reported the new ExM methodology using swellable hydrogels assembled by click-chemistry-based non-radical linking of two complementary tetrahedral monomers (“tetra-gel” ExM). It allowed visualization of herpes simplex virus type 1 (HSV-1) virions with substantially smaller spatial errors (deviation of 9.2 nm) compared with classical PAAG-expanded virions (14.3 nm) [95]. Importantly, tetra-gel ExM much better preserves the spherical shapes of the virions compared with polyacrylamide-gel ExM. Obviously, the new protocol can also be applied to a variety of other enveloped viruses including SARS-CoV-2.

## Figures and Tables

**Figure 1 ijms-22-06558-f001:**
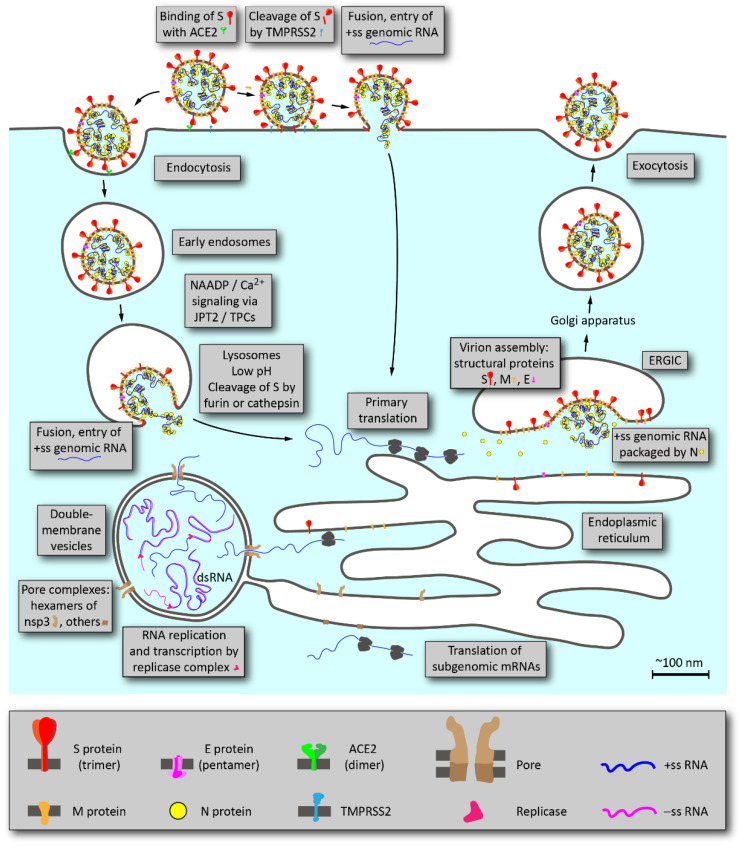
Schematic outline of SARS-CoV-2 life cycle. Proteins and membrane structures are drawn approximately to scale. In the graphical legend at the bottom, elements are magnified 4-fold.

**Figure 2 ijms-22-06558-f002:**
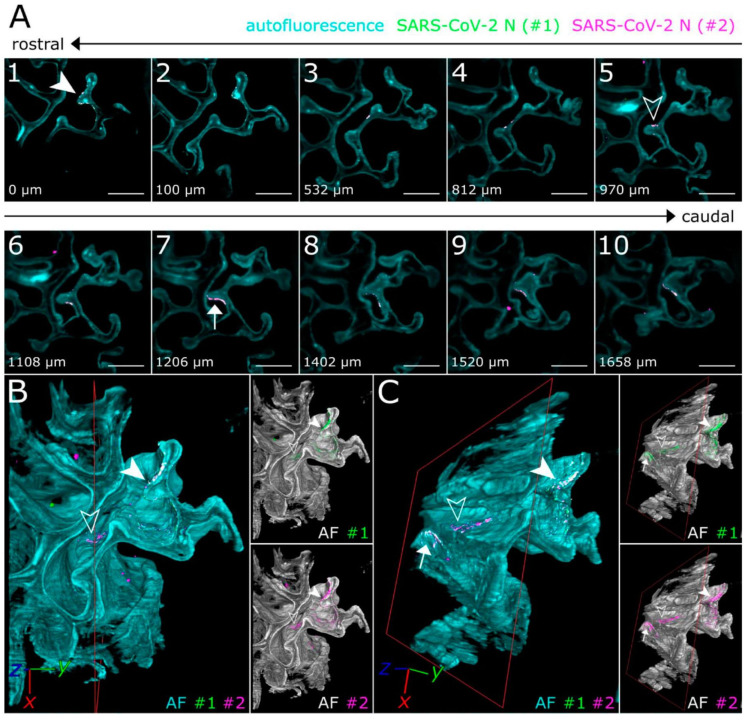
Light-sheet 3D fluorescence imaging of SARS-CoV-2-infected ferret nasal turbinates. Cyan—autofluorescence (overall tissue structure); green and magenta—staining with two different antibodies against N-protein. Only overlapping green and magenta signals are considered to be specific for SARS-CoV-2-infected cells (marked by arrows). (**A**) Individual optical sections. (**B**,**C**) 3D reconstructions. Images from Zaeck et al. [26].

**Figure 3 ijms-22-06558-f003:**
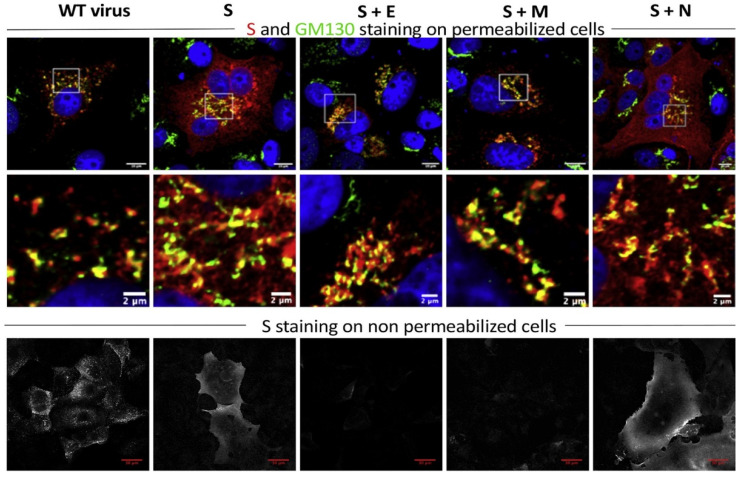
Studying intracellular distribution of spike protein by immunocytochemistry and confocal microscopy. Cells were either infected with SARS-CoV-2 (left column) or transfected with plasmids encoding S, E, M and/or N proteins as indicated at the top. GM130 (green) was used as a cis-Golgi marker. Note that S protein (red in the top panels or white in the bottom panels) are exported on the plasma membrane when expressed alone or with N protein, but it retains into intracellular vesicles when coexpressed with M or E proteins. Images from Boson et al. [36].

**Figure 4 ijms-22-06558-f004:**
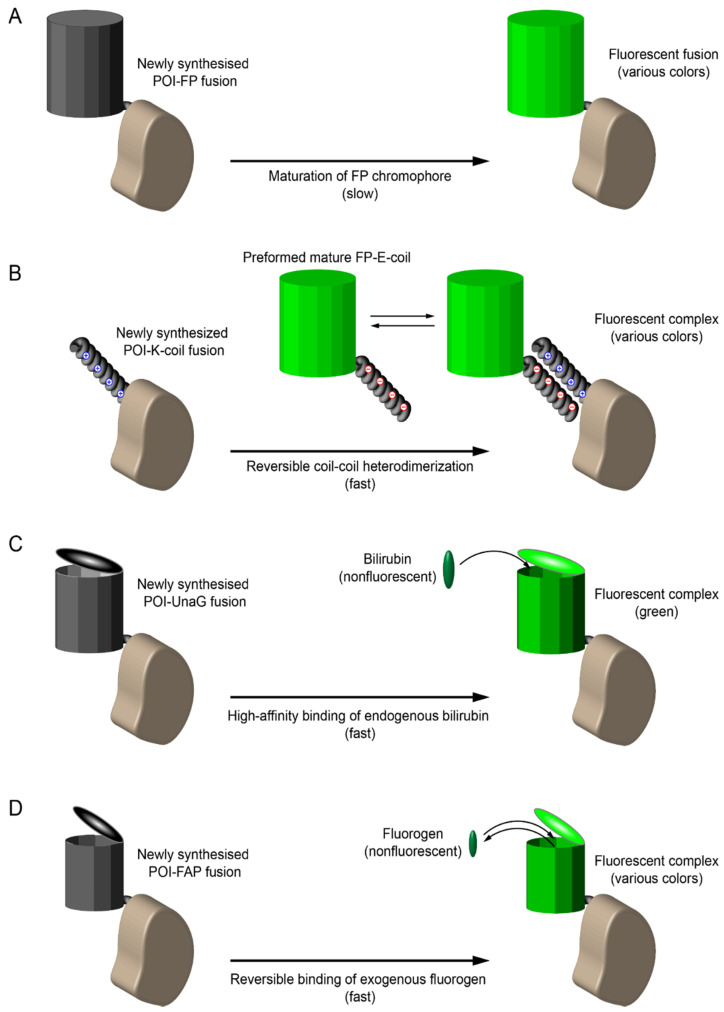
Different strategies of genetic tagging. Protein of interest (POI, brown) can be fused with various tags to ensure fluorescence labeling in live cells. (**A**) Classical labeling with GFP-like fluorescent proteins (FP). Chromophore maturation in FP is a relatively slow process (tens of minutes), thus early events of POI trafficking and interactions are missed. This problem can be solved by immediate labeling of newly synthesized POI with tags depicted in **B**–**D**. (**B**) KECs labeling. A heterodimerizing pair of coils (positively charged K-coil and negatively charged E-coil) is used to attract preformed mature FP to nascent POI. It can be applied to POIs with distinct intracellular localization (fibers, membranes, etc.) to distinguish between target signal and diffuse FP background. (**C**) Labeling with bilirubin-binding fluorescent protein UnaG. (**D**) Labeling with fluorogen-activating proteins (FAP), which bind exogenously added fluorogenic dyes.

**Table 1 ijms-22-06558-t001:** Summary of fluorescent probes and imaging methods used to study SARS-CoV-2.

Applications	Probes	Imaging Methods	References
Viral entry in cell lines	ACE2 fused with FP	Confocal and epifluorescence microscopy	[16,17,18,20,22]
	VSV fused with FP	Spinning disk confocal microscopy, flow cytometry	[21]
	Cell filaments fused with FP	Confocal and TIRF microscopy	[22]
	Immunofluorescence	Confocal microscopy	[16,17,22]
	FD-curve-based AFM guided by fluorescence	Atomic force microscopy, fluorescence microscopy	[18]
Viral entry in tissues	Immunofluorescence	Confocal microscopy	[23,24,25]
	Immunofluorescence	Light-sheet microscopy, confocal microscopy	[26]
Endocytosis of virus	Immunofluorescence	Confocal microscopy	[27,29,31]
	ACE2 fused with FP (pseudoviruses)	Confocal microscopy	[28,29]
	TPC1 and TPS2 fused with FP	Spinning disk confocal microscopy	[31]
Viral RNA release	3a/nsp3 fused with FP	Confocal microscopy, cryo-EM	[32,34]
	S, M, N fused with FP	Wide-field and confocal microscopy	[35,46]
	Staining with fluorescent dyes	Confocal microscopy	[32]
	Immunofluorescence	Confocal microscopy	[33,36]
Influence of viral proteins on the host cell	Immunofluorescence	Confocal microscopy	[38,41,44]
	3a, E, 7a, nsp1, Rip3, Gal3, TFEB and LAMP1 fused with FP	3a—confocal microscopy and cryo-EM; the rest—confocal microscopy	[39,40,42,43,44]
RNA tracking	N protein with Cy5, ssRNA with 6-FAM	Confocal microscopy, photobleaching	[46]
	FISH, immunofluorescence	Confocal microscopy	[46,47,48,49,50,51]
SARS-CoV-2 drug screening and inhibitor testing	Immunofluorescence	Fluorescence microscopy	[52]
	Quantum dots	Single-molecule imaging	[53]
	FRET-based reporter for TMPRSS2 detection	Microplate reader	[54]
	FlipGFP-based reporters for 3CLpro or PLpro detection	Fluorescence microscopy	[55,56]
	Fluorescence dye labeling	TIRF microscopy	[58]
Recombinant SARS-CoV-2 expressing reporter genes	S, Rab5, PTS1, LAMP1 tagged with FP	Confocal microscopy	[60]
	SARS-CoV-2 tagged with an FP	Confocal microscopy	[59,61,62,65]

## Data Availability

Data sharing not applicable.

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
