# Peer review of "Studying SARS-CoV-2 with Fluorescence Microscopy"

_ijms, 2021, doi:10.3390/ijms22126558_

Round 1

Reviewer 1 Report

This review is an interesting source of information but, there are some missing parts that will be useful for the readers.

For example there are not figures associate with the described technic, please add some.

Also there are missing key references:

Zaeck LM, Scheibner D, Sehl J, Müller M, Hoffmann D, Beer M, Abdelwhab EM, Mettenleiter TC, Breithaupt A, Finke S. Light Sheet Microscopy-Assisted 3D Analysis of SARS-CoV-2 Infection in the Respiratory Tract of the Ferret Model. Viruses. 2021 Mar 23;13(3):529. doi: 10.3390/v13030529. PMID: 33807059; PMCID: PMC8004956.

Ogando NS, Dalebout TJ, Zevenhoven-Dobbe JC, Limpens RWAL, van der Meer Y, Caly L, Druce J, de Vries JJC, Kikkert M, Bárcena M, Sidorov I, Snijder EJ. SARS-coronavirus-2 replication in Vero E6 cells: replication kinetics, rapid adaptation and cytopathology. J Gen Virol. 2020 Sep;101(9):925-940. doi: 10.1099/jgv.0.001453. PMID: 32568027; PMCID: PMC7654748.

Arista-Romero M, Pujals S, Albertazzi L. Towards a Quantitative Single Particle Characterization by Super Resolution Microscopy: From Virus Structures to Antivirals Design. Front Bioeng Biotechnol. 2021 Mar 26;9:647874. doi: 10.3389/fbioe.2021.647874. PMID: 33842446; PMCID: PMC8033170.

Lv J, Wang Z, Qu Y, Zhu H, Zhu Q, Tong W, Bao L, Lv Q, Cong J, Li D, Deng W, Yu P, Song J, Tong WM, Liu J, Liu Y, Qin C, Huang B. Distinct uptake, amplification, and release of SARS-CoV-2 by M1 and M2 alveolar macrophages. Cell Discov. 2021 Apr 13;7(1):24. doi: 10.1038/s41421-021-00258-1. PMID: 33850112; PMCID: PMC8043100.

Cortese M, Laketa V. Advanced microscopy technologies enable rapid response to SARS-CoV-2 pandemic. Cell Microbiol. 2021 Feb 17:e13319. doi: 10.1111/cmi.13319. Epub ahead of print. PMID: 33595881; PMCID: PMC7995000.

Author Response

Response to Reviewer 1 Comments

This review is an interesting source of information but, there are some missing parts that will be useful for the readers.

For example there are not figures associate with the described technic, please add some.

Response: 

In fact, the manuscript contains a figure (Fig. 2 in the original version, Fig. 4 in the revised version) that illustrates some recently described techniques for fluorescence protein labeling. Other methods such as immunostaining and FISH are old and very well known; we believe that there is no need to give a Figure for them.    

Also there are missing key references:

Zaeck LM, Scheibner D, Sehl J, Müller M, Hoffmann D, Beer M, Abdelwhab EM, Mettenleiter TC, Breithaupt A, Finke S. Light Sheet Microscopy-Assisted 3D Analysis of SARS-CoV-2 Infection in the Respiratory Tract of the Ferret Model. Viruses. 2021 Mar 23;13(3):529. doi: 10.3390/v13030529. PMID: 33807059; PMCID: PMC8004956.

Ogando NS, Dalebout TJ, Zevenhoven-Dobbe JC, Limpens RWAL, van der Meer Y, Caly L, Druce J, de Vries JJC, Kikkert M, Bárcena M, Sidorov I, Snijder EJ. SARS-coronavirus-2 replication in Vero E6 cells: replication kinetics, rapid adaptation and cytopathology. J Gen Virol. 2020 Sep;101(9):925-940. doi: 10.1099/jgv.0.001453. PMID: 32568027; PMCID: PMC7654748.

Arista-Romero M, Pujals S, Albertazzi L. Towards a Quantitative Single Particle Characterization by Super Resolution Microscopy: From Virus Structures to Antivirals Design. Front Bioeng Biotechnol. 2021 Mar 26;9:647874. doi: 10.3389/fbioe.2021.647874. PMID: 33842446; PMCID: PMC8033170.

Lv J, Wang Z, Qu Y, Zhu H, Zhu Q, Tong W, Bao L, Lv Q, Cong J, Li D, Deng W, Yu P, Song J, Tong WM, Liu J, Liu Y, Qin C, Huang B. Distinct uptake, amplification, and release of SARS-CoV-2 by M1 and M2 alveolar macrophages. Cell Discov. 2021 Apr 13;7(1):24. doi: 10.1038/s41421-021-00258-1. PMID: 33850112; PMCID: PMC8043100.

Cortese M, Laketa V. Advanced microscopy technologies enable rapid response to SARS-CoV-2 pandemic. Cell Microbiol. 2021 Feb 17:e13319. doi: 10.1111/cmi.13319. Epub ahead of print. PMID: 33595881; PMCID: PMC7995000.

Response:

Thanks a lot for drawing our attention to these works! In the revised manuscript, we discussed and cited these papers (refs 31, 52, 87, 24, 15, respectively).

Reviewer 2 Report

In this review, the authors present various studies on the different steps of SARS-CoV-2 infection, mainly using fluorescence microscopy, but also cryo-EM and electron microscopy. This review provides a comprehensive overview of studies from the early stages of viral infection to viral maturation, and is considered to be an important review not only for fluorescence microscopy but also for virology.

Analysis of the structure of the Spike protein is of interest in terms of binding of neutralizing antibodies and enhancement of infectivity of the variants. Structural analysis using cryo-EM is a powerful tool for revealing protein structures and has produced many results in the structural analysis of Spike proteins. Although the subject of this paper is fluorescence microscopy, this review will provide an overview of the SARS-CoV-2 research by discussing the results of Spike structural analysis using cryo-EM.

Author Response

Response to Reviewer 2 Comments 

In this review, the authors present various studies on the different steps of SARS-CoV-2 infection, mainly using fluorescence microscopy, but also cryo-EM and electron microscopy. This review provides a comprehensive overview of studies from the early stages of viral infection to viral maturation, and is considered to be an important review not only for fluorescence microscopy but also for virology.

Analysis of the structure of the Spike protein is of interest in terms of binding of neutralizing antibodies and enhancement of infectivity of the variants. Structural analysis using cryo-EM is a powerful tool for revealing protein structures and has produced many results in the structural analysis of Spike proteins. Although the subject of this paper is fluorescence microscopy, this review will provide an overview of the SARS-CoV-2 research by discussing the results of Spike structural analysis using cryo-EM.

Response:

We added the reference on cryo-EM study of antibody-Spike interactions:

“Using cryo-EM, Schoof et al. demonstrated that a single domain antibody Nb6 stabilizes Spike protein in a conformation where its binding domain is inaccessible for interactions with ACE2 [19].”

19. Schoof, M.; Faust, B.; Saunders, R.A.; Sangwan, S.; Rezelj, V.; Hoppe, N.; Boone, M.; Billesbølle, C.B.; Puchades, C.; Azumaya, C.M.; et al. An Ultrapotent Synthetic Nanobody Neutralizes SARS-CoV-2 by Stabilizing Inactive Spike. Science 2020, 370, 1473–1479.

Reviewer 3 Report

The manuscript “Studying SARS-CoV-2 with fluorescence microscopy” reviews the SARS-CoV-2 life cycle and gives pointers for fluorescent microscopy as a mean to study the biology and some tools for anti-viral drug screening. The manuscript is collection of useful information, however, there is certainly a scope for improvement.

  1. It is strongly encouraged to incorporate some IF images from already published articles or authors can provide their own research images. A review article without this seem to do injustice to the title and content.
  2. Please organize the content into tables for life cycle (genes, proteins and its function etc) and fluorescence microscopy based studies of viral life cycle sections.
  3. A table can also be made by listing different fluorescence microscopy techniques and its application in SARS-CoV-2 studies.

Author Response

Response to Reviewer 3 Comments 

The manuscript “Studying SARS-CoV-2 with fluorescence microscopy” reviews the SARS-CoV-2 life cycle and gives pointers for fluorescent microscopy as a mean to study the biology and some tools for anti-viral drug screening. The manuscript is collection of useful information, however, there is certainly a scope for improvement.

1. It is strongly encouraged to incorporate some IF images from already published articles or authors can provide their own research images. A review article without this seem to do injustice to the title and content.

Response:

Two new Figures (Figs. 2 and 3) that show some nice examples of fluorescence imaging of SARS-CoV-2 were added to the revised manuscript. 

2. Please organize the content into tables for life cycle (genes, proteins and its function etc) and fluorescence microscopy based studies of viral life cycle sections.

3. A table can also be made by listing different fluorescence microscopy techniques and its application in SARS-CoV-2 studies.

Response:

We added a table (Table 1) that provides a summary of fluorescent probes and imaging methods used to study different stages of SARS-CoV-2 life cycle, its proteins and gRNA, and drug screening.